# Role of Vaginal Mucosa, Host Immunity and Microbiota in Vulvovaginal Candidiasis

**DOI:** 10.3390/pathogens11060618

**Published:** 2022-05-25

**Authors:** Subatrra Nair Balakrishnan, Haizat Yamang, Michael C. Lorenz, Shu Yih Chew, Leslie Thian Lung Than

**Affiliations:** 1Department of Medical Microbiology, Faculty of Medicine and Health Sciences, University Putra Malaysia, Serdang 43300, Selangor, Malaysia; subatrranair@gmail.com (S.N.B.); haizatyamang@gmail.com (H.Y.); 2Department of Microbiology and Molecular Genetics, University of Texas McGovern Medical School, Houston, TX 77030, USA; michael.lorenz@uth.tmc.edu

**Keywords:** host immune response, pattern recognition receptors (PRRs), vulvovaginal candidiasis, vaginal mucosa, vaginal microbiota

## Abstract

Vulvovaginal candidiasis (VVC) is a prevalent gynaecological disease characterised by vaginal wall inflammation that is caused by *Candida* species. VVC impacts almost three-quarters of all women throughout their reproductive years. As the vaginal mucosa is the first point of contact with microbes, vaginal epithelial cells are the first line of defence against opportunistic *Candida* infection by providing a physical barrier and mounting immunological responses. The mechanisms of defence against this infection are displayed through the rapid shedding of epithelial cells, the presence of pattern recognition receptors, and the release of inflammatory cytokines. The bacterial microbiota within the mucosal layer presents another form of defence mechanism within the vagina through acidic pH regulation, the release of antifungal peptides and physiological control against dysbiosis. The significant role of the microbiota in maintaining vaginal health promotes its application as one of the potential treatment modalities against VVC with the hope of alleviating the burden of VVC, especially the recurrent disease. This review discusses and summarises current progress in understanding the role of vaginal mucosa and host immunity upon infection, together with the function of vaginal microbiota in VVC.

## 1. Introduction

Vulvovaginal candidiasis (VVC) is a mucosal infection at the vaginal tissue and the vulva of the lower female reproductive tract, which is caused predominantly by *Candida albicans*, an opportunistic fungal pathogen. Over 90% of VVC cases are caused by *C. albicans* and the remaining 10% are caused by the non-*albicans Candida* (NAC) species, such as *C. glabrata*, *C. krusei*, *C. tropicalis* and *C. parapsilosis* [1]. Ascending genital tract infections and their association with direct and indirect economic costs make VVC a global health concern [2]. Seventy-five percent of women are reported to experience this vaginal yeast infection at least once in their lifespan [3]. Following anaerobic bacterial vaginosis, candidiasis is the second most prevalent root cause of vaginal inflammation that is associated with symptoms, such as itching, irritation, dysuria, dyspareunia, cottage cheese-like vaginal discharges, or soreness of the vulva that may persist for days or even weeks which can be aggravated by sexual intercourse [4]. In addition, the normal human vaginal microenvironment is more acidic (pH of 3.8–4.5) compared to other mammals (pH of 5.4–7.8), with Lactobacilli as the dominant organisms in the human vaginal microbiota—more than 70% of the human vaginal microbiota in some women [5]. Diagnosis of VVC is made by direct microscopy from vaginal swabs or smears, in which yeast cells, pseudohyphae and hyphae can be observed. An increase in vaginal pH of about 4.5, erythema of the vaginal mucosa, vulval oedema, and a thick white discharge are also common signs of VVC [6].

Recurrent VVC (RVVC) occurs if a woman experiences more than three episodes within 12 months. The majority of infections in women with RVVC are caused by *C. albicans* [7]. Women with RVVC have been identified as having predisposing genetic factors or hypersensitivity to *Candida* that make them susceptible to this infection [8]. According to Foxman et al. (2013), based on their study in five European countries and the United States, a global prevalence of 3871 RVVC cases per 100,000 women have been documented with the highest frequency (9%) observed in women aged between 25 and 34 years old [9]. Moreover, VVC and RVVC tend to infect women primarily during their reproductive years, as they experience an elevated endogenous oestrogen level [10]. In addition, there are two types of risk factors for VVC, namely host factors and behavioural factors. Pregnancy, menopausal hormone replacement (MHR), uncontrolled diabetes mellitus, immunosuppression, antibiotic and glucocorticoid usage as well as genetic effects are host-related risk factors. Meanwhile, contraceptive usage, personal hygiene practices and sexual behaviour fall under the behavioural risk factors for VVC [11].

*Candida* possesses a plethora of virulence factors which include the ability to evade host defence, adherence to the surface epithelium, biofilm formation and invasion by the secretion of extracellular enzymes [12]. *Candida* adherence to host surfaces is essential for the primary colonisation of human tissues, which is a prerequisite for infections. A key mechanism in the onset of VVC is the adherence of *Candida* to vaginal epithelial cells (VEC) [13], via adhesins, which are unique cell surface proteins that play their role by mediating *Candida* spp. cell adhesion to biotic and abiotic surfaces by cell surface physicochemical features [14,15]. They also contribute to biofilm development on host or abiotic surfaces, which is linked to a majority of infections caused by *Candida* spp. and contribute to treatment failures, as these structures are more resistant to antifungal drugs [16,17]. Biofilm formation on implanted medical devices (e.g., vascular or urinary catheters, artificial heart valves, dental implants, etc.) of *Candida*, especially in *C. albicans*, has made it one of the most common causes of nosocomial infections, which are becoming extremely hard to treat due to both inherent and acquired antifungal resistance [18]. Moreover, infection-induced production of hydrolytic enzymes improves organism adherence, invasion and elimination of immunological components in the host, and has a role in nutrition acquisition. Hydrolytic enzymes, such as secreted aspartyl proteinases (SAP), phospholipases (PL) and candidialysin [19] play a role in VVC pathogenesis. A recent in vitro study carried out by Ali et al. (2018) reported that the deletion of *SAP1*–*SAP3* genes has reduced the *C. albicans* vaginopathic capacity [20]. The expression of *SAP1*–*SAP3* genes has been associated with vaginal infection that may implicate the yeast-to-hyphal switching in the virulence of *C. albicans* [21]. The *SAP1*–*SAP3* genes are the first *SAP* genes to be expressed by *C. albicans* during the development of lesions in VVC [22]. In addition, Roselletti et al. (2017) reported that *SAP1* and *SAP3* genes were expressed in almost half of the symptomatic women, while *SAP2* expression was elevated in all of the patients [23]. According to Pericolini et al. (2015), *SAP2* expression has also led to an increase in caspase-1 expression in murine and human VEC [24]. On the other hand, candidalysin is required for the activation of immunopathological signalling in the vaginal mucosa. The ability of candidalysin to destroy epithelia and potentially promote free vaginal heparin sulphate may contribute to the fungal fitness strategy to protect against PMN-mediated clearance at the mucosal surface [25]. 

The vaginal mucosa is the most significant and adaptable structure in the female reproductive system and is the surface upon which *Candida* species adhere to initiate infections, whilst vaginal immunity confers protection to the host upon VVC occurrence. In addition, the composition of the vaginal microbiota also plays a significant role in the onset of VVC. This review summarises the role of three crucial components: the vaginal mucosa, vaginal immunity and the vaginal microbiota in VVC.

## 2. Vaginal Mucosa 

### 2.1. Vaginal Structure 

The vagina is an elastic and muscular organ of the female genital tract that connects the uterus and cervix to the outside of the body (Figure 1A). An outermost mucosa layer, the interstitial muscularis layer, and the deeper adventitia layer are the three different layers that make up the vaginal wall structure (Figure 1B). The mucosal layer is a structure that continues from the uterine lining and can be divided into two sublayers; the first sublayer is the epithelial layer which rests on an underlying second layer of lamina propria. The vaginal epithelial layer or vaginal epithelium (VE) is a multi-layered stratified squamous structure which undergoes differentiation into several distinct and parallel layers known as strata [26]. The uppermost layer of VE is also known as the stratum corneum and is closest to the lumen. It undergoes rapid shredding and regeneration throughout the menstruation cycle [27]. The exfoliation of VE cells serves as the initial defence mechanism of the vagina as this action removes potential pathogenic *Candida* bound to it [28].

### 2.2. VEC and Its Significance 

As the first cell lining to be in contact with *Candida*, the apical VEC layer plays an important role as the protective barrier and the first line of defence. This apical VEC layer together with the other underlying VEC layers is held together by intercellular transmembrane proteins which include F11R, desmosomal adhesion proteins (JAM3), tight junction protein 1 (TJP1), claudin-1 and E-cadherin. These junction proteins are found occupying the multi-layered structure of the VE that assists in maintaining the overall cellular integrity of the layer and simultaneously restricts the dissemination of *Candida* [29]. In fact, it was proposed that the mechanism by which *C. albicans* invades mucosal epithelial tissue is mediated through proteolytic degradation of adherence junctions [30,31], although other routes of entry, including candidalysin-mediated epithelial lysis, have also been suggested. The VEC also contributes significantly to the activation of innate immunity through expressed pattern recognition receptors (PRRs) on the apical surface of the cells. PRRs are capable of detecting the presence of pathogenic *Candida* upon binding (ligation) with pathogen-associated molecular patterns (PAMPs) secreted by or presented on the surface of the pathogen [19]. Previous research revealed various PRRs are responsible for *Candida* recognition, dominated by the Toll-like receptors (TLRs), C-type lectin receptors (CLRs), the nucleotide-binding oligomerisation domain (NOD)-like receptor (NLR) and the retinoic acid-inducible gene I (RIG-I) -like receptors (RLRs) [32]. Although each receptor portrays different functions, TLRs are known to be expressed within VEC, which mediate the release of antimicrobial peptides (AMP), pro-inflammatory and anti-inflammatory cytokines, along with co-stimulatory molecules to promote adaptive immunity [33]. Pro-inflammatory cytokines, such as TNF-α, IL-1β, and IL-6, are released by VEC, and this promotes the migration of local immune cells to affected sites. The release of these cytokines activates and mediates the proliferation of innate immune cells, primarily macrophages and neutrophils, to the affected site [34]. A study conducted by Diletta et al. (2020) shows that, upon exposure to *C. albicans* hyphae, significantly higher production of TNF-α among RVVC patients was observed compared with the control group (healthy individuals), indicating its possible role in acute local immune response [35]. Anti-inflammatory cytokines (IL-10 and TGF-β) are also released by VEC and are believed to regulate inflammatory responses, which prevent excessive damage to local cells [36]. This dynamic release of cytokines gives an impact on the balance of vaginal response toward the pathogen. 

Pathogen recognition by PRRs activates NF-κB and MAPK cascades leading to the production of AMPs by VEC [37], including lysozyme, lactoferrin, serine leukocyte protease inhibitor (SLPI), calprotectin, and elafin as well as α- and β-defensins throughout the vaginal tract that exhibits a wide range of innate immunity, including antifungal properties [38,39]. Calprotectin, a metal-binding protein, was proven to be an effective antifungal agent as it induces iron starvation and consequently inhibits iron availability to *C. albicans* [40]. Research on the antifungal effect of SLPI shows cytoplasmic deflation of *C. albicans* upon treatment with SLPI peptides, suggesting a penetrative effect of SLPI through the cell wall of the fungus [41]. Meanwhile, Chairatan et al. (2016) found that α-defensin exhibits antifungal activities by blocking the adhesion of *C. albicans* within the epithelial cells of human intestines and inhibits its biofilm formation [42]. An in vivo study using a mouse model of VVC caused by *C. albicans* revealed a significant increase in murine β-defensin 1 expression during early infection (acute phase), paired with polymorphonuclear leukocyte (PMN; neutrophils) infiltration, demonstrating its role in first-line defence against fungal infection and as inducers of innate inflammatory mediators [43]. 

Host-pathogen interplay within the mucosa is undeniably unique, considering the ability of VEC to discriminate between disease-causing microbes and commensal. As VEC is exposed to diverse microorganisms within the mucosa interface, it is crucial to understand the tolerance mechanism of VEC towards different types of organisms. In a review, Naren et al. (2010) suggested that microbial homeostasis within mucosa is highly influenced by the nature of the resident microbes [44]. As we know, *Candida* is a benign part of the normal vaginal microbiota in many women. However, due to vaginal dysbiosis, it increases the fungal burden within the vagina, triggers yeast-to-hyphal switching and releases virulence factors of *C. albicans* [19]. The virulence factors of *Candida*, together with the presence of PAMPs, make them recognisable as a “threat” and thus, they stimulate inflammatory immune responses [44]. In addition, another review by McLure et al. (2014) suggested that the differential responses are affected by the cellular localisation of TLRs. Unlike commensals, pathogenic microorganisms can breach the epithelial barrier and invade the deeper layer of mucosa, resulting in a pro-inflammatory response that can precipitate the pathology associated with VVC [37]. 

## 3. Vaginal Host Immunity 

The initial line of defence against *Candida* infections at most body sites involves the innate immune system. The cells of various epithelial surfaces coordinate the innate response to counteract proliferating pathogens by recruiting additional cellular mediators, like neutrophils, and inducing the expression of AMPs. The adaptive immune response is defined by specificity, and it plays a role in pathogen removal in the late stages of infection and in the development of immunological memory [45]. The vaginal mucosa, both its tissue layers together with the cervicovaginal fluids, contain both innate and acquired immune responses. The antifungal activity mediated by epithelial cells has been proven to be both fungistatic and non-inflammatory, implying that it functions as a basic defence mechanism to prevent excessive tissue damage and to maintain asymptomatic commensalism, perhaps via broad-specificity AMPs [46]. Adaptive immunity, in contrast, engenders pathogen-specific defence mechanisms where the antigen-presenting cells (APC) digest the pathogen antigens and deliver them to T cells, resulting in T-cell activation. Following that, the activation of cytokine production and antibody synthesis will take place where B cells will produce antibodies and the T cells will mediate the protective mechanism [47]. 

### 3.1. Role of PRRs in Immune Recognition 

Generally, the innate immune responses are mediated by the interplay of the innate immune cells of the host that express PRRs on their surface together with the PAMPs of the pathogens [48]. PRRs expressed by immune cells can be classified into four different classes, such as Toll-like receptors (TLRs), C-type lectin receptors (CLRs), the nucleotide-binding oligomerisation domain (NOD)-like receptor (NLR) and the retinoic acid-inducible gene I (RIG-I) -like receptors (RLRs) [1]. In general, monocytes and neutrophils in the circulation, as well as the macrophages in affected tissues, are the major cells of the host’s innate immune response that are involved in identifying invading pathogens [48]. Neutrophils are the main immune cells involved in the regulation of VVC and, thus, neutropenic women are more susceptible to this infection [49,50]. In support of that statement, a study using the murine VVC model by Ibrahim et al. (2013) has reported that vaccination with rAls3p-N protein of *C. albicans*, formulated with alum adjuvant (NDV-3), elicited protection against VVC by which anti-rAls3p-N antibodies work as an opsonin to increase IFN-γ primed neutrophil activity in killing *C. albicans* [51]. TLRs and CLRs families are the most common receptors that identify *Candida*-associated molecular patterns which are mostly expressed by monocytes on their cell membranes. While in neutrophils, TLRs expression is minimal, whereas phagocytic receptors, such as complement receptor 3 (CR3) and Fcγ receptors (FcγRs), are highly expressed [52]. TLRs can be classified into cell-membrane associated receptors (TLR1, TLR2, TLR4, TLR5 and TLR6) or intracellular receptors (TLR3, TLR7, TLR8 and TLR9). TLRs mainly recognise fungal components; for instance, TLR2 is involved in the identification of phospholipomannan, TLR4 is involved in the recognition of O-linked mannans, TLR6 is involved in the recognition of zymosan (essentially a crude mixture of fungal cell wall components), while TLR9 is involved in the detection of fungal DNA [53]. On the other hand, CLRs are responsible for identifying polysaccharide structures from *C. albicans*, i.e., Dectin-1 detects β-glucans, whereas the macrophage mannose receptor (MR) and dendritic cell-specific ICAM-3-grabbing nonintegrin (DC-SIGN) as well as Dectin-2 recognise the N-linked mannans [52]. During infection, the detection of PAMPs by PRRs can induce gene expression in the inflammatory responses by initiating the production of pro-inflammatory cytokines, type I interferons (IFNs), chemokines together with antimicrobial proteins, and the proteins that participated in PRRs signalling modulation [54].

### 3.2. Role of PAMPs in Immune Recognition 

Fungal PAMPs are conserved and constitute primarily cell wall carbohydrate structures specific to these microbes. In *C. albicans*, the outer cell wall consists of proteins glycosylated with O-linked and N-linked and phosphorylated mannans that eventually prevent the immune recognition of the inner cell wall components. This layer is thinner at the level of the bud scar caused by yeast division, allowing PAMPs from the inner cell wall to be recognised by the immune system [55]. *Candida* species often mask the inner cell wall layer, such as β-glucan and chitin, from recognition by Dectin-1, one of the well-characterised CLRs and other PRRs [56]. Therefore, recognition of PAMPs by the innate immune system allows the distinction between ‘self’ and the microbial ‘non-self’ [57]. Neutrophils have been demonstrated to be able to create extracellular traps and trigger the cell wall remodelling in *Candida*, which leads to better β-glucan exposure and Dectin-1 binding [58]. The majority of fungal pathogens, including *C. albicans*, have a core structure of β-(1,3)-glucan covalently bonded to β-(1,6)-glucan and chitin (a β-(1,4)-linked polymer of N-acetylglucosamine (GlcNAc)) [59]. β-glucans are polymeric carbohydrates that are disclosed to regulate inflammatory responses in both in vitro and in vivo models [60]. The degree of branching, polymer length and tertiary structure of β-glucans impact their immunomodulatory effects, but no consensus has been achieved on the basic structural requirements for biological activity, and various forms of glucans have distinct biological effects [61]. When Dectin-1 binds to β-glucan, an Src-Syk-CARD9-dependent pathway is initiated, which results in the triggering of the NF-κB and the transcription of antifungal cytokines, such as IL-6, IL-1β, along with IL-23 [58]. Dectin-1 stimulates cytokine production in a synergistic manner with TLR2 and TLR4, but it also generates IL-1β, IL-17 and IL-10 independently of these TLRs. The activation of signalling pathways within dendritic cells (DCs), as a result of PRR activation, leads to the initiation of a specific adaptive cellular immune response. Dendritic cells are found throughout the vaginal and cervical epithelia, where they adhere to *Candida* and initiate cell differentiation together with activation of cell signals that cause inflammation, cell proliferation and induce adaptive immunity [62]. The cells of the innate immune system detect portions of the *C. albicans* cell wall skeletal and matrix components. The complicated interaction between host receptors and *C. albicans* cell wall components, together with the released compounds, can activate various inflammasomes [63]. Fungal cell wall components can cause inflammation by activating epithelial and immunological receptors, resulting in the release of pro-inflammatory cytokines and chemokines. However, pathogenic fungi can mask these cell wall components to remain undetected [64].

### 3.3. Antimicrobial Defence Mechanism against C. albicans

The cationic AMPs of innate immunity have received a lot of interest in the last two decades as prospective candidates for developing new antimicrobials [65]. Direct targeting of AMPs to microbial membranes and subsequent disruption suggests that resistance is less likely to be developed, which is critical in the fight against antifungal-resistant vaginal infections. The reasons put forth for this theory are because AMPs are more likely to be cidal, as opposed to static, and they kill cells quickly through mechanisms that are hard to bypass, such as membrane disruption. AMPs often possess several targets, such as the cell membrane, specific proteins and nucleic acids, hence, even if resistance to one develops, others remain to take the task forward [66]. For instance, beyond AMPs binding to membranes, there is evidence that they can adhere to target ribosomal subunits, inhibit DNA, RNA, protein, and macromolecule synthesis, as well as interfere with respiration, intracellular protein folding, and iron regulation [67]. These natural peptides are used by species throughout evolution as a first line of defence [68]. These AMPs are produced by professional phagocytes, as well as epithelial cells of the skin and the respiratory, gastrointestinal, and genitourinary tracts [69]. Human β-defensins have broad antifungal activity and chemotactic functions and are some of the AMPs expressed within the VE [70]. The fungicidal activity of the human β-defensins is mainly displayed through membrane permeabilisation, resulting in intracellular ion leakage, which inhibits planktonic and biofilm growth alike [71,72]. Complementing their direct antifungal effects, β-defensins also exhibit immunomodulatory function via pro-inflammatory cytokines stimulation and immune cell recruitment [73]. The cathelicidin LL-37 (CRAMP in mice), is a linear cationic alpha-helical peptide produced by various epithelial cells, including those of the gut as well as those in the squamous epithelium of the cervix and vagina, as well as in cervicovaginal secretions, and is one of the best-studied AMPs in human [74]. LL-37 associates with the cell wall and cell membrane where in *C. albicans*, the cell wall β-1,3-exoglucanase Xog1 has been identified as an LL-37 receptor. The Xog1–LL-37 interactions lead to cell wall remodelling that results in elevation of Xog1 enzyme activity, thus lowering the *C. albicans* adhesion [75]. Levinson et al. (2009) reported that increased levels of LL-37 and a few other AMPs were discovered in the vaginal fluids of women with various persistent vaginal infections, including VVC [76]. In addition, AMPs have also been found to bind to the cell wall glucan, potentially interfering with cell wall functions, including adhesion [77]. A study by Woodburn et al. (2019) have used designed antimicrobial peptides (dAMP) i.e., RP504, RP544, RP556, and RP557 and evaluated them on fluconazole-sensitive and resistant *C. albicans*, *C. glabrata*, *C. tropicalis*, and *C. parapsilosis*, as well as inherently resistant *C. krusei*, where these AMPs were reported to be fungicidal since the minimum inhibitory concentration (MIC) values were similar to their minimum fungicidal concentration (MFC) values. Combined with activity in a rodent VVC model, the findings support the clinical evaluation of dAMPs for topical treatment of VCC and recurrent VVC infections [78]. 

### 3.4. Adaptive Immune Responses upon VVC Occurrence

Adaptive immunity includes the fungus-specific defence mechanisms that are developed directly or indirectly through cell-mediated immunity (T cells). Most of the vaginal T cells are thought to move to the VE as a result of local antigenic stimulation and/or inflammatory chemokines [62]. Since mucosal candidiasis is more common in people who have T cell immunosuppression, therefore some experimental models have shown that T cells can protect humans against *Candida* infections and the decrease in T cell-mediated immunity has been linked to increased susceptibility to VVC in women who are immunocompromised [79]. Among HIV-infected women, lower CD4^+^ T cell counts and higher HIV viral loads were associated with VVC [80]. The isolation and detection of antibodies in vaginal washings have been used to demonstrate humoral immunity in the vaginal mucosa [81]. In response to pathogens, B cells and immunoglobulin-secreting plasma cells were found to move into the vaginal epithelium [38]. IgG and IgA secreting plasma cells are abundant in the endocervix lamina propria but limited in the vagina, indicating that immune microenvironments exist in the female genital tract [82]. TGF-β, a well-known anti-inflammatory cytokine, has been found to be expressed constitutively in the mucosal surfaces of the vaginal mucosa regardless of the existence of a Th1 specific anti-*Candida* response [83]. Secretion of CCL20 and β-defensin 2 by VEC attract CCR6-expressing dendritic cells to the mucosa, where they process fungal antigens and activate Th responses, including Th17 cells. In conjunction with IL-1β and IL-6, TGF-β may also trigger Th17 differentiation where the Th17 cell production of IL-17 enhances neutrophil activity, while IL-22 strengthens epithelial barrier function [84]. These released cytokines induce the expression of anti-*Candida* peptides and, hence, it has been suggested that they have the potential to halt the development of VVC [85]. There is increasing evidence suggesting that IL-17A and IL-17F-mediated immunity may play a role in host defence in the mouse vagina [86]. According to Peter et al. (2020) induction of a strong IL-17-related gene signature in the vagina during estrogen-dependent murine VVC has also been observed [87], but how consequential this signalling is in preventing the symptoms of VVC remains under debate.

## 4. Vaginal Microbiota 

### 4.1. Vaginal Microbiota Plays a Predominant Role in VVC Occurrence 

Apart from its unique structure, the human vagina is colonised by numerous distinct microorganisms, collectively known as vaginal microbiota and which can encompass microorganisms with potential commensal, symbiotic or pathogenic relationships with the host [88]. The mucosal layer of the vagina is predominantly occupied by *Lactobacillus* species [89], facultative anaerobic, Gram-positive, catalase-negative, non spore-forming, rod-shaped bacteria. They rely on VEC for resources. They can utilise glycogen produced by endocervical and fallopian tube cells, using α-amylase to convert it into glucose, then fermenting that to produce lactic acid that reduces the vaginal pH [89,90]. Acidic pH inhibits the yeast-to-hyphal transition in *C. albicans* and thus maintains the fungus in a less-immunostimulatory state. The quantity of cell-free glycogen available in vaginal discharge is likely to be a significant influence on their growth and acid generation that makes them the dominant microflora of the human vagina [91]. They are also vital for vaginal health, and a reduction in their population is known to increase the risk of vaginal infections [92]. 

Some of the most common *Lactobacillus* species found are *L. crispatus, L. plantarum, L. gasseri, L. jensenii, L. fermentum, L. brevis, L. iners, L. casei, L delbrueckii, L. vaginalis,* and *L. salivarius* [89,93]. Among reproductive-age women, there can be up to 10 different species of *Lactobacillus* found within the epithelium layer of the vagina. Ravel et al. (2011) studied the vaginal microbiota profile of asymptomatic women from different ethnicities and divided them into five different clusters. These clusters are referred to as community state types (CST) that reflect the dominance of microbiota in each individual. They found that most of the women belong to the four clusters of CSTs (*Lactobacillus*-dominated). These microbial communities are classified as CST I, CST II, CST III and CST V which are dominantly occupied by *L. crispatus, L. gasseri, L. iners*, and *L. jensenii*, respectively. CST IV, on the other hand, is a diverse and polymicrobial which is dominated by other types of anaerobic bacteria, including *Aerococcus, Mobiluncus, Atopobium, Megasphaera, Peptoniphilus, Sneathia, Gardnerella, Eggerthella, Prevotella, Finegoldia*, together with *Dialister* [94]. All of these microorganisms make up the vaginal microbiota and colonise in a dynamic environment [95]. Though each woman exhibits a different vaginal microbiota community, those with *Lactobacillus* as the dominant member are more often associated with healthy vaginal homeostasis [96]. 

Fungal communities are also a part of the vaginal microbiota that exists within the mucosa layer and the microflora of its species are known as vaginal mycobiota [97]. The vaginal mycobiota of women is mainly dominated by *Candida* species, together with a minimal representation of other fungal species, including *Cladosporium, Eurotium* and *Alternaria* genera [98]. *C. albicans* exists in the urogenital tract of healthy women asymptomatically and exists within eubiotic communities in the vaginal microbiota [97]. However, it is one of the leading causes of vaginitis in humans, by far the most common fungal cause, and the differential immunogenicity of the yeast and hyphal forms can trigger symptomatic VVC [97,98]. A wide range of contributing factors, such as quorum sensing between cells, nutrient starvation, vaginal pH, vaginal dysbiosis and the presence of N-acetylglucosamine (GlcNAc) can promote the morphological transitions and pathogenicity of *C. albicans* [18,99]. This illustrates a combination of behavioural (sexual practices, antibiotic use, hormonal contraception) and innate factors (immune responses, genetic conditions) that may also influence the overgrowth of *C. albicans* in vaginal mucosa [100]. 

### 4.2. Interaction between Lactobacillus and Candida 

The vaginal microbiota plays a crucial role in shielding the lower genital tract of women against *Candida* invasion. The bacterial microbiota within the vagina provides the first-line barrier protection as it creates competition for space and resources for *Candida* and, hence, prevents them from invading the vaginal epithelia [19]. Moreover, some *Lactobacillus* spp. exhibit specific and direct antifungal effects against *Candida*. For instance, *L. rhamnosus* GR-1 and *L. reuteri* RC-14 were found to exhibit antagonistic effects against an NCAC species, the vaginal pathogen *C. glabrata* [101]. Additionally, Li et al. (2019) demonstrated that both *L. crispatus* and *L. delbrueckii* inhibited 60% to 70% of *C. albicans* in a VVC Sprague-Dawley rat model compared with non-treated controls [102]. Production of lactic acid by *Lactobacillus* is proven to inhibit the hyphal formation of *C. albicans* [103]. A study by Camile Nina et al. (2019) revealed that lactic acid, in combination with major peptidoglycan hydrolase released by *L. rhamnosus*, GG successfully inhibits the hyphal formation of *C. albicans* [104]. Paniágua et al. (2021) also reported that *C. albicans’* hyphae development was significantly hampered by *L. casei* Shirota [105]. *C. albicans* was found to modulate low pH into an alkaline condition to undergo hyphal formation, thus, in a way, supporting the idea that vaginal acidic condition is an inhibitory mechanism against the morphological transition in *C. albicans* [106]. Lactic acid also stimulates the release of anti-inflammatory cytokines. Hearps et al. (2017) found that lactic acid induces anti-inflammatory cytokine IL-1RA release and suppresses pro-inflammatory mediators production by cervicovaginal epithelial cell lines [107]. However, not only with lactic acid; it was also proven that *Lactobacillus* conditioned supernatant (LCS), isolated from *L. fermentum* and *L. crispatus*, was revealed to downregulate the expression of hypha-related genes of *C. albicans*, including *ALS3, ECE1, SAP5* and *HWP1* [103]. 

*Lactobacillus* also produces antimicrobial metabolites, such as hydrogen peroxide and biosurfactants, which are highly effective against *Candida* [108]. It was found that hydrogen peroxide-producing *Lactobacillus*, like *L. acidophilus*, was able to inhibit the growth of *C. albicans* compared with the non-hydrogen peroxide-producing strain [109]. However, *C. albicans* is quite resistant to ROS, so the antifungal activity might be due to overlapping mechanisms contributed by *Lactobacillus* [110]. On the other hand, Prisicila et al. (2020) reported that biosurfactants produced by *L. crispatus* were able to reduce the adhesion of *C. albicans* to HeLA cells with an inhibitory effect on biofilm formation [111]. Biosurfactants produced by *L. gasseri* together with *L. jensenii* were also able to suppress biofilms produced by *C. albicans, C. tropicalis,* and *C. krusei* by 25% to 35% [112]. A study by Chew et al. (2015) found that cell-free supernatants (CFS) of *L. rhamnosus* GR-1 and *L. reuteri* RC-14 were able to inhibit biofilm formation and suppress biofilm-related genes (EPA6 and YAK1) of *C. glabrata* [113].

Figure 2 below shows the series of events that take place during the interaction of *C. albicans* with *Lactobacillus* spp. As illustrated, upon infection and formation of *Candida* biofilm on vaginal mucosae, the normal microflora that are present (e.g., *Lactobacillus*) regulate themselves to produce metabolites such as lactic acid to inhibit *Candida* overgrowth via acidification of the vaginal mucosae. The metabolites produced by *Lactobacillus* inhibit *C. albicans* colonisation, either by preventing adherence to the epithelial cell wall or by exerting a fungistatic effect due to the surge in concentrations of organic acids. Adherence of *Candida* species is prevented by the co-aggregation action and saturation of adhesion sites by lactobacilli. After the interaction, the presence of lactobacilli suppresses the expression of virulence genes of *C. albicans*, namely *ALS3*, *EFG1* and *HWP1* genes that are related to hyphal production and adhesion. A recent study by Wang et al. (2017) reported that *L. crispatus* CFS significantly down-regulates the expression of hyphae-specific genes *ALS3* (0.140-fold), *HWP1* (0.075-fold), and *ECE1* (0.045-fold), while it up-regulates the expression of the negative transcriptional regulator gene *NRG1* with (1.911-fold) [114]. In the event of *Candida* colonisation, *Lactobacillus* can change the host immunological response, attract granulocytes and promote immune defence.

The composition of the vaginal microbiota naturally dominates the mucosal layer in a commensal and balanced ecosystem. However, several external stimuli and intrinsic factors can cause disruption to the ecosystem of the vaginal microbiota. Factors, such as sexual hormonal changes, douching, menstruation, sexual activity, menopause, stress, pregnancy, lactation, antibiotics usage and oral contraceptive pills may result in dysbiosis in the vaginal microbiota [89,115]. This is characterised by the depletion of healthy vaginal microflora causing overgrowth of dysbiosis-associated pathogens [116]. As the vaginal microflora is commonly dominated by *Lactobacillus*, dysbiosis-induced neutralisation of vaginal pH favours *Candida* overgrowth by diminishing the microbiota-derived protective mechanism within the vaginal mucosa, allowing the morphogenesis and adherence of pathogenic *Candida* to the VE [117]. Such dysbiosis has been shown to contribute to vulnerability towards vaginal infection and the possibility of developing VVC [116,118,119]. 

## 5. Future Perspectives 

One of the promising treatments in VVC intervention is vaginal microbiota transplantation (VMT). This method involves the administration of microbiota from a healthy individual to the vagina of VVC patients as replacement therapy, restoring the imbalance microbiota community [120]. The continuous study of VMT is largely influenced by the success of faecal microbiota transplantation (FMT) in treating gastrointestinal diseases [121]. The effectiveness of FMT has been established through several clinical trials, including treatment for irritable bowel syndrome and *Clostridioides difficile* infection (previously known as *Clostridium difficile* infection) that show significant alleviation of symptoms, with fewer adverse side effects and an improvement of life quality among patients [122,123]. A recent study by Iriana et al. (2020) reported that FMT targeting gut microbiota in ulcerative colitis patients shows an increase in anti-*Candida* antibodies relative to placebo controls [124]. Because a *Lactobacillus*-dominated microbiota has been considered indicative of vaginal health, further study on its potential as a prophylactic agent or adjuvant in women with gynaecological diseases is warranted. A study by Lev-Sagie et al. (2019) shows that VMT in patients with bacterial vaginosis shifted the microbiota profile closer to that of the healthy donors [125]. The therapeutic effects of probiotics can be extended to cases of candidiasis. Xie et al. (2017) concluded that patients treated with exogenous probiotics exhibit a better short-term clinical cure and mycological cure rates when prescribed together with conventional antifungal therapy [126]. This correlates with a study conducted by Kovachev et al. (2015), as they found that patients who were administered probiotics in combination with azole treatment showed a significant reduction in clinical complaints, an improvement in vaginal microbiota and a healthier vaginal environment [127]. This research on VMT provides a platform to apply VMT not only as a sole treatment but as a supporting intervention in treating VVC.

## 6. Conclusions

In conclusion, the vaginal mucosa is the primary contact point with microorganisms in VVC and therefore it plays a primary role in the development of symptomatic infections as a physical barrier, by triggering immunological responses, and as the host of a surface-associated community of potentially antagonistic bacteria. The detection of PRRs by PAMPs contributes to the immune recognition upon VVC occurrence. The vaginal microbiota within the mucosal layer enhances the defence mechanism of the vagina through acidic pH regulation, the release of AMPs and physiological control against vaginal dysbiosis. The significant role of vaginal microbiota in maintaining vaginal health also promotes its application, i.e., vaginal microbiota transplantation (VMT) as a potential adjuvant treatment to alleviate, especially the recurrent form of VVC, alongside the use of probiotics and their metabolites. Hence, a further understanding of the vaginal microenvironment and host-pathogen interactions against *Candida* is crucial to initiate and sustain the long-term development of improved VVC intervention.

## Figures and Tables

**Figure 1 pathogens-11-00618-f001:**
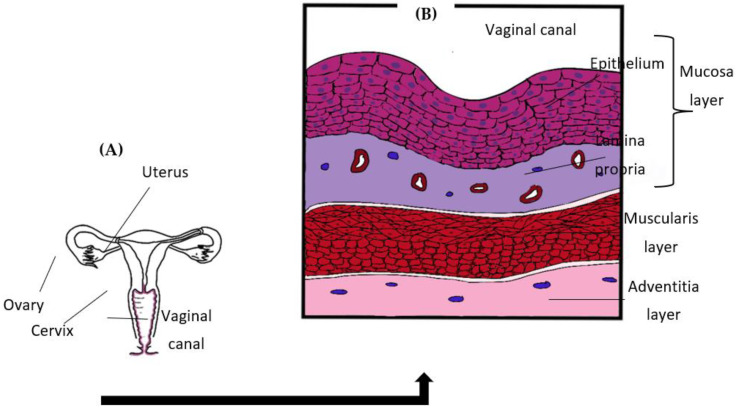
(**A**) Female genital tract. (**B**) Illustration of the vaginal wall structure.

**Figure 2 pathogens-11-00618-f002:**
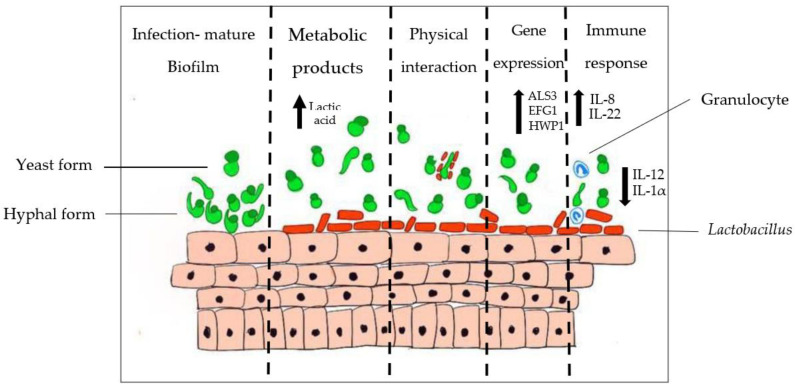
Interactions between *C. albicans* and *Lactobacillus* spp.

## Data Availability

Not applicable.

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
