# Peer review of "Role of Vaginal Mucosa, Host Immunity and Microbiota in Vulvovaginal Candidiasis"

_pathogens, 2022, doi:10.3390/pathogens11060618_

Round 1

Reviewer 1 Report

The present review presents some interesting aspects in relation to the pathogenesis of CVV, however the material included in the work does not agree with the title because the mechanisms of protection are not correctly addressed. In this regard, it shows some conceptual errors and omissions. In the opinion of this reviewer, the most interesting aspect is the interaction of Lactobacilli with the fungus and the transplantation of microbiota.

In general, the manuscript needs a thorough revision by the authors as it contains several repeated paragraphs in the different sections, where information is repeated. The different sections should be edited as a whole, which will benefit the quality of the paper. Also the order in which the concepts are presented AND the concepts included in each section.

It is also noticeable the errors in relation to the writing and naming of the microorganisms where for example and in relation to Candida albicans, that once defined the genus and species should be presented in abbreviated form, it should also be written in italics. The same happens for the denomination of lactobacillus.

In relation to the role of immunological mechanisms, some conceptual errors and omissions are observed.

Finally, the paper would be enriched if the authors included some details of the works cited and not only the enumeration of their conclusions from the reviews consulted by the authors.

Specific comments:

Introduction

- Line 25 . the fungal infection is not only in the lumen, the fungus infects the tissue. Please correct

- Line 50-52: The prevalence  for of species varies with geographic regions, therefore this concept should be revised.

- Line 52-56: The pathology classification should be revised, the authors cite a 2004 paper. The advance of knowledge has led to discard this classification.

- Line 62-64: review the expression "women who..." is not a  virulence factor.

- Line 62-68: the redaction of the whole paragraph requires revision.

- Line 72: pathogen denomination must be cheeked.

Vaginal Histology

    Throughout the section the microorganisms are not correctly identified.

   The paragraph should be shortened as it presents redundant information.

 In Figure 1, what does vaginal microenvironment mean? The figure shows a histological slice with partial description of tissue characteristics.

Vaginal Mucosae and Its Significance….

Line 106-109: should be included in the histologic description section.

Line 110-113: This paragraph is redundant, these concepts have already been presented.  The paper needs to be edited as a whole to avoid unnecessary repetition.

Line 122-125: this paragraph is also repeated.

Line 127: identification of the microorganism is incorrect.

Line: 129-148: this section should be revised and completed. At the local level, vaginal epithelial cells produce pro- and anti-inflammatory cytokines important in the regulation of the local microenvironment, the authors should include this information. This paragraph also omits information on innate lymphocytes.

Line150-151: The function of vagina mucosa against Candida species can also be demonstrated through the role of caspase-associated recruitment domain 9 (CARD9) as an adaptor protein in PRRs regulation system.

This  sentence is incorrect. It is not clear to this reviewer because information on CARD-9 is included in this section. The papers cited correspond to the description of this C-type lectin receptor adaptor, which defines the importance in antifungal defense and not particularly at the vaginal level.  In this sense, other genetic defects in innate receptors are directly associated with susceptibility to CVV. 

Line 159-162: same comment.

Vaginal Microbiota Plays a Predominant Role…

 Several paragraphs are redundant with the previous sections. A thorough reading is needed to eliminate so many repeated paragraphs. The name of the microorganisms should also be revised.

Suggested to include details of some of the papers cited and not just the description of the conclusions.

Interaction between Lactobacillus and Candida

 In this section the name of any microorganism does not appear in italics.  What are the metabolites released by the Lactobacilli that inhibit the adherence of the fungus?  It would be useful to have more information on this subject.

Figure 2 does not illustrate this concept. Figure 2 instead shows some adhesins of the fungus without any information in the text. These molecules are well identified and cloned and could be referenced here.  Although the authors include this information in another section below, the figure shows elements that have not been presented. The information provided here is very superficial.

Line 258-271: None of the authors mentioned have the reference number.

 Pathogenesis of VVC

In this section the authors do not describe the pathogenesis of the disease, but present some of the virulence factors of the fungus.

The paragraph on biofilm should be shortened and the authors should focus on the existing evidence in both animal models and patients on this virulence factor in the pathology they are addressing and the discussion about it.

Line 315-318: what is the reference of the work of Ali et al. It would be very interesting to know more about this work.

What are the other authors that provide evidence on Sap1-3? No reference is included in the text. There are very important papers related to Sap and CVV that should be included.

Host Immune Response against VVC

Line 320-337 : in this section the authors include in a very general way the role of VEC, as part of the innate immune response, and very general concepts about the initiation of the adaptive response that could well be associated to the characteristic of the response for this particular niche.

Role of PRRs in Immune Recognition

  In this section it would be very important to include some receptors of the mentioned families that are associated with susceptibility to infection, which are also mentioned in the review, which would allow the reader to be better informed.

 Line 344: RPPs do not act after infection, please correct.

 Line 366-369: What is the point of including NK cells here? This reference was already cited above in relation to the secretory function of this population.  The work included is from 2002, is there any other more current reference that talks about the importance of this population in CVV? Why do the authors prioritize this population and not the innate lymphoid cells, for example?

In this section there  are appear for first time abbreviations of elements that have been mentioned before.

Adaptive Immune Responses upon VVC Occurrence

 Line 394-397: the authors again list the cell populations that have already been mentioned more than 3 times. Again, the authors focus on NK cells. What is the interest in this population based on?

Line 400-408: in this paragraph the authors describe mechanisms of innate immunity, while in the title they refer to adaptive immunity. This paragraph should be placed when the authors refer to the pathogenesis of CVV.

Line 409-413: this paragraph should include papers related to the current discussion on the role of LITh17 in CVV.

Author Response

Comments to the Author:

The present review presents some interesting aspects in relation to the pathogenesis of CVV, however the material included in the work does not agree with the title because the mechanisms of protection are not correctly addressed. In this regard, it shows some conceptual errors and omissions. In the opinion of this reviewer, the most interesting aspect is the interaction of lactobacilli with the fungus and the transplantation of microbiota. In general, the manuscript needs a thorough revision by the authors as it contains several repeated paragraphs in the different sections, where information is repeated. The different sections should be edited as a whole, which will benefit the quality of the paper. Also, the order in which the concepts are presented AND the concepts included in each section. It is also noticeable the errors in relation to the writing and naming of the microorganisms where for example and in relation to Candida albicans, that once defined the genus and species should be presented in abbreviated form, it should also be written in italics. The same happens for the denomination of Lactobacillus. In relation to the role of immunological mechanisms, some conceptual errors and omissions are observed. Finally, the paper would be enriched if the authors included some details of the works cited and not only the enumeration of their conclusions from the reviews consulted by the authors.

We appreciate the valuable and constructive comments provided by the REVIEWER 1. We have taken them fully into account in our revised manuscript. We feel the changes have strengthened the paper, which we hope is now suitable for publication. Thank you.

  1. Line 25: the fungal infection is not only in the lumen; the fungus infects the tissue. Please correct

Response: Thank you. We have replaced the word lumen.

  1. Line 50-52: The prevalence for of species varies with geographic regions, therefore this

concept should be revised.

Response: Thank you. The sentence has been revised. Please refer to line from 52- 55.

  1. Line 52-56: The pathology classification should be revised; the authors cite a 2004 paper. The advance of knowledge has led to discard this classification.

Response: Thank you for your comment. We have now revised the sentences accordingly.

  1. Line 62-64: review the expression "women who..." is not a virulence factor.

Response: Thank you for the correction. We have now revised the sentence and also the whole paragraph on virulence factors. Please refer to line from 62-84. 

  1. Line 62-68: the redaction of the whole paragraph requires revision.

Response: Thank you. We have revised the whole paragraph and please refer to line from 57-61.

  1. Line 72: pathogen denomination must be checked.

Response: Thank you. The pathogen denomination has been changed.

Vaginal histology

  1. Throughout the section the microorganisms are not correctly identified. The paragraph should be shortened as it presents redundant information.

In Figure 1, what does vaginal microenvironment mean? The figure shows a histological slice with partial description of tissue characteristics.

Response: Thank you for the comment. The paragraph has been shortened and the redundant information have been removed. The word “vaginal microenvironment’ has been removed and the figure has also been explained in detail. 

  1. Line 106-109: should be included in the histologic description section

Response: Thank you. The information from line 106-109 has now been incorporated in the histologic description section (Line 103-107).

  1. Line 110-113: This paragraph is redundant; these concepts have already been presented.  The paper needs to be edited as a whole to avoid unnecessary repetition.

Response: Thank you. The sentence has been removed. We have also edited the manuscript as a whole to remove repetition.

  1. Line 122-125: this paragraph is also repeated.

Response: Thank you. The paragraph has been removed.

  1. Line 127: identification of the microorganism is incorrect.

Response: Thank you for the correction. The microorganism’s name has been corrected.

  1. Line: 129-148: this section should be revised and completed. At the local level, vaginal epithelial cells produce pro- and anti-inflammatory cytokines important in the regulation of the local microenvironment, the authors should include this information. This paragraph also omits information on innate lymphocytes.

Response: Thank you for your comment. The information has been included and kindly refer to the line 128-136.

  1. Line150-151: The function of vagina mucosa against Candida species can also be demonstrated through the role of caspase-associated recruitment domain 9 (CARD9) as an adaptor protein in PRRs regulation system.

Response: Thank you. This part has been removed and incorporated into Section 3.1 (Line 211-223)

  1. Line 159-162: same comment.

Response: Thank you. This part has been removed and incorporated into Section 3.1 (Line 211-223)

  1. Several paragraphs are redundant with the previous sections. A thorough reading is needed to eliminate so many repeated paragraphs. The name of the microorganisms should also be revised. Suggested to include details of some of the papers cited and not just the description of the conclusions.

Response: Thank you for the valuable suggestions given. The redundant portions have been removed and the organisms name have been corrected accordingly.

  1. In this section the name of any microorganism does not appear in italics. What are the metabolites released by the lactobacilli that inhibit the adherence of the fungus?  It would be useful to have more information on this subject.

Response: Thank you. Trivial names are often used as a simplified way of naming a bacterial genus. A trivial name should neither be written with capital first letter nor in italic. Examples of trivial names are: lactobacilli, mycobacteria, salmonella, staphylococci and streptococci therefore the word lactobacilli did not been italicized. However, when refer to a specific microbial species, a trivial name referring to a complete genus should never be used.

Reference:
https://www.vetbact.org/popup/popup.php?id=59#:~:text=Trivial%20names%20are%20often%20used,%2C%20salmonella%2C%20staphylococci%20and%20streptococci.

  1. Figure 2 does not illustrate this concept. Figure 2 instead shows some adhesins of the fungus without any information in the text. These molecules are well identified and cloned and could be referenced here. Although the authors include this information in another section below, the figure shows elements that have not been presented. The information provided here is very superficial.

Response: Thank you. The description for Figure 2 has been modified and please refer to line 403-416.

  1. Line 258-271: None of the authors mentioned have the reference number.

Response: Thank you. The reference has been sorted.

  1. In this section the authors do not describe the pathogenesis of the disease, but present some of the virulence factors of the fungus.

Response: Thank you. This part has been removed and some of the information has been added under introduction. Please refer to line 62-84.

  1. The paragraph on biofilm should be shortened and the authors should focus on the existing evidence in both animal models and patients on this virulence factor in the pathology they are addressing and the discussion about it.

Response: Thank you. The paragraph on biofilm has been removed and the information regarding the biofilm formation as a virulence factor has been covered under introduction. Please refer to line from 65-68.

  1. Line 315-318: what is the reference of the work of Ali et al. It would be very interesting to know more about this work.

Response: Thank you.  The reference has been added and more information about the study has been added in line 79-81.  

  1. What are the other authors that provide evidence on Sap1-3? No reference is included in the text. There are very important papers related to Sap and CVV that should be included.

Response: Thank you. The information om SAP has been included in line 81-84.

  1. Line 320-337: in this section the authors include in a very general way the role of VEC, as part of the innate immune response, and very general concepts about the initiation of the adaptive response that could well be associated to the characteristic of the response for this particular niche.

Response: Thank you. The paragraph for the role of VEC has been modified and more information has been added under the Section 2.2 (Line 116-164).

  1. In this section it would be very important to include some receptors of the mentioned families that are associated with susceptibility to infection, which are also mentioned in the review, which would allow the reader to be better informed.

Response: Thank you for the suggestion. The example of receptors has been included in line 200-205.

  1. Line 344: RPPs do not act after infection, please correct.

Response: Thank you for the correction. The words “upon infection” has been replaced with the words “during infection”.

  1. Line 366-369: What is the point of including NK cells here? This reference was already cited above in relation to the secretory function of this population.  The work included is from 2002, is there any other more current reference that talks about the importance of this population in CVV? Why do the authors prioritize this population and not the innate lymphoid cells, for example?

Response: Thank you for the suggestion. The paragraph now has been altered and it covers other innate immune cells as well. (Line 189-194)

  1. In this section there are appear for first time abbreviations of elements that have been mentioned before.

Response: Thank you for the correction. The first-time abbreviations have been addressed and corrected accordingly.

  1. Line 394-397: the authors again list the cell populations that have already been mentioned more than 3 times. Again, the authors focus on NK cells. What is the interest in this population based on?

Response: Thank you for the comments. We have now also covered discussion of other immune cells.

  1. Line 400-408: in this paragraph the authors describe mechanisms of innate immunity, while in the title they refer to adaptive immunity. This paragraph should be placed when the authors refer to the pathogenesis of CVV.

Response: Thank you for the correction. The adaptive immunity paragraph has been revised and more related information has been added. Please refer to line 296-318.

  1. Line 409-413: this paragraph should include papers related to the current discussion on the role of LITh17 in CVV.

Response: Thank you for the suggestion. The role of Th17 in VVC have been included now in line 313-318.

Reviewer 2 Report

This ms deals with pathomechanisms of the VVC and recurrent VVC in relation to vaginal microbiota & innate immunity of the vaginal mucosa. The authors use phrases not convergent with those found in medical articles on VVC but also not used in medical microbiology like as microbiota or make it consistently throughout the entire text. Paragraphs are not fully integrated and in consequence there are multiple redundances and repetitions. Future perspectives are rather are not supported with substantial data on effectiveness of the new therapies including vaginal microbiota replacement which is not the same as FMT directed toward a highly specific spore-forming pathogen uncomparable to biology Candida in vagina, although FMT seems to decrease Candida in gut which is not mentioned by the authors. Conclusions are not conclusive but reflect authors good wishes. 

There are multiple errors in the text. Here are some examples:

  • line 42: which criteria for VVC were adopted?
  • line 56: what is hormone replacement?
  • line 65: why unmanageable?
  • line 79: why modified?
  • line 95 & 121: there no sound data on glycogen as energy source for lactobacilli; this is a very old hypothesis
  • line 169: There is no longer a single genus Lactobacillus; we have now a new taxonomy of the lactobacilli
  • line 217: hydrogen peroxide is not produced by lactobacilli in anaerobic environment of the vagina
  • line 232: it is not true since Candida prefers also as low pH as lactobacilli
  • line 253: what does it mean: lactobacilli suppress virulence genes of Candida? Do authors can provide more data? (and citations!)     

Author Response

Comments to the Author:

This ms deals with pathomechanisms of the VVC and recurrent VVC in relation to vaginal microbiota & innate immunity of the vaginal mucosa. The authors use phrases not convergent with those found in medical articles on VVC but also not used in medical microbiology like as microbiota or make it consistently throughout the entire text. Paragraphs are not fully integrated and in consequence there are multiple redundances and repetitions. Future perspectives are rather are not supported with substantial data on effectiveness of the new therapies including vaginal microbiota replacement which is not the same as FMT directed toward a highly specific spore-forming pathogen uncomparable to biology Candida in vagina, although FMT seems to decrease Candida in gut which is not mentioned by the authors. Conclusions are not conclusive but reflect authors good wishes. 

We appreciate the valuable and constructive comments provided by the REVIEWER 2. We have taken them fully into account in our revised manuscript. We feel the changes made have strengthened the paper, which we hope it is not more suitable for publication. Thank you.

  1. Line 42: which criteria for VVC were adopted?

Response: Thank you. The criteria for VVC adopted were addressed in line 42-46. 

  1. line 56: what is hormone replacement?

Response: Thank you. The whole sentence regarding the classification of the RVVC have been removed.

  1. line 65: why unmanageable?

Response: Thank you. The word “unmanageable” has been replaced with “uncontrolled”.

  1. line 79: why modified?

Response: The justification has been added into at line 423-426.

  1. line 95 & 121: there no sound data on glycogen as energy source for lactobacilli; this is a very old hypothesis

Response: Thank you. The sentence has been removed.

  1. line 169: There is no longer a single genus Lactobacillus; we have now a new taxonomy of the lactobacilli

Response: Thank you for the comment. The sentence has been removed.

  1. line 217: hydrogen peroxide is not produced by lactobacilli in anaerobic environment of the vagina.

Response: Thank you for the comment. The sentence has been removed.

  1. line 232: it is not true since Candida prefers also as low pH as lactobacilli

Response: Thank you for the comment. Kindly refer to line 384-386.

  1. line 253: what does it mean: lactobacilli suppress virulence genes of Candida? Do authors can provide more data? (and citations!)     

Response: Thank you for the comment. Related study has been discussed and cited (Line 411-416).

Round 2

Reviewer 1 Report

 Rev comments in the attached letter

Author Response

Reviewer 1:

The new version of the manuscript shows a significant improvement. The paper has been re-edited and redundant paragraphs removed. Comments on cited articles have been included. However, since the present review is intended to provide a current perspective on the subject, I request the authors to include some relevant citations related to this pathology.

1)         The fungus infects the tissue. Remove the word “wall”

Response: Thank you. The word “wall” has been replaced with tissue (Line 26).

2)         REV: in relation with SAP role during VVC, the authors must read and include the following works

Roselletti, E.et al 2017. NLRP3 inflammasome is a key player in human vulvovaginal disease caused by Candida albicans. Sci. Rep.

Pericolini, E.et al 2015. Secretory Aspartyl Proteinases Cause Vaginitis and Can Mediate Vaginitis Caused by Candida albicans in Mice. mBio

Response: Thank you for the suggestions. The suggested literatures have been included in the discussion and cited accordingly. Please refer to Line 85-88.

3)         Line 86-88: review punctuation marks.

Response: Thank you. This paragraph has been rephrased accordingly. Please refer to Line 90-94.

4)         Vaginal histology

The quality of the work has improved considerably, although this point needs to be reinforced. The figure is still very poor and so is the image quality. The authors give a description of the tissue, however the sections they refer to are neither included nor represented in the image.

“An internal mucosal layer, interstitial muscularis layer, together with the external adventitial layer are the three different layers that make up the vaginal wall histology (Figure 1B).

It is difficult for the reader to associate the text and the image. I recommend replacing it with a better, illustrative image that includes what is referred to in the text. A more descriptive caption would help for a better understanding. For example, magnification, staining, section, etc.

Response: Thank you for your comment. We have now reworked the image as suggested to better explain the description in the text (Figure 1)

5)         In line 123 Change IL-1 alpha for IL-1 beta.

Response: Thank you. IL-1 alpha has been changed to IL-1 beta (Line 139)

6)         REV: This section has been significantly improved. It would be interesting for the authors to read and include a recent paper on Beta defensins and VVC.

Miró MS, et al 2021. Candida albicans Modulates Murine and Human Beta Defensin-1 during Vaginitis. J Fungi.

Response: Thank you. The suggested literature has been included in the discussion and cited accordingly. Please refer to Line 155-159.

7)         In this section, please check the reference 37. Is a journal or book???

37-Junko, “Novel Mechanism behind the Immunopathogenesis of Vulvovaginal Candidiasis: ‘Neutrophil Anergy,’” pp. 1–12, 569 2018.

Response: Thank you. Reference [37] (now Reference [43]) is a journal article and it

has now been amended accordingly.

8)         I agree with the modification but this sentence is incorrect and need be corrected: The function of vagina mucosa against Candida species can also be demonstrated through the role of caspase-associated recruitment domain 9 (CARD9) as an adaptor protein in PRRs regulation system. The function of CARD-9 has not been studied in the vaginal mucosa.

Response: Thank you, this is a fair point. The whole paragraph discussing about CARD9 has now been removed.

9)         Line 216: Please include TNF- alpha

            Line 259: Include IL-1beta

            Line 266: remove “antimicrobial peptide”, only used the abbreviation.

Response: Thank you. We have amended them accordingly.

10)       REV: Please include these references, they are key when referring to SAP and VVC

Roselletti, E et al 2017. NLRP3 inflammasome is a key player in human vulvovaginal disease caused by Candida albicans. Sci. Rep.

Pericolini, E.et al 2015. Secretory Aspartyl Proteinases Cause Vaginitis and Can Mediate Vaginitis Caused by Candida albicans in Mice. mBio

Response: Thank you for the suggestions. The suggested literatures have been included in the discussion and cited accordingly. Please refer to Line 85-88.

11)       Line 281, please include one comment about Beta defensin in VVC.

Response: Thank you. The information on β-defensins in VVC has now been added. Please refer to Line 280-285.

12)       Line 298, use the abbreviation VE

            Line 314: correct IL-1 beta

Response: Thank you. We have amended them accordingly.

13)       While the role of Lth17 in gut is better elucidated, current research questions the role of this population during CVV. Please read and include at least one paragraph regarding the role in candida vaginitis.

- Li, J. et al. 2017 Mucocutaneous IL-17 immunity in mice and humans: Host defense vs. excessive inflammation. Mucosal Immunol.

- Okada, S. et al. 2016 Chronic mucocutaneous candidiasis disease associated with inborn errors of IL-17 immunity. Clin. Transl. Immunol.

- Peters, B.M. et al. 2019. The Interleukin (IL) 17R/IL-22R Signaling Axis Is Dispensable for Vulvovaginal Candidiasis Regardless of Estrogen Status. J. Infect. Dis.

Response: Thank you. The paragraph regarding the role of IL-17 in Candida vaginitis has now been added. Please refer to Line 322-326.

14)       Conclusion

Line 473: It would be more correct to refer to the recurrent form of mycosis.

Response: Thank you. The term VVC has now been replaced with recurrent form of mycosis.

Reviewer 2 Report

In spite of the authors' response there are still uncorrected phrases and sentences to be changed or rejected. Example: hormon replacement should be menopausal hormone replacement (MHR) etc.      

Author Response

In spite of the authors' response there are still uncorrected phrases and sentences to be changed or rejected. Example: hormone replacement should be menopausal hormone replacement (MHR) etc.

Response: Thank you. The word hormone replacement has now been replaced with menopausal hormone replacement (MHR). Please refer to Line 57.

The manuscript has been carefully proofread and revised. We believe that the contents and the clarity of our manuscript are much improved after the revision.